# Meta-Analysis of the Response of the Productivity of Different Crops to Parameters and Processes in Soil Nitrogen Cycle under Biochar Addition

Leiyi Zhang [1,2], Zhuohao Wu [1], Jingyan Zhou [1], Lingli Zhou [1], Yang Lu [1], Yangzhou Xiang [3], Renduo Zhang [2], Qi Deng [4],* and Wencheng Wu [1],*

1   South China Institute of Environmental Sciences, Ministry of Ecology and Environment, Guangzhou 510655, China
2   Guangdong Provincial Key Laboratory of Environmental Pollution Control and Remediation Technology, School of Environmental Science and Engineering, Sun Yat-sen University, Guangzhou 510006, China
3   School of Geography and Resources, Guizhou Education University, Guiyang 550003, China
4   Key Laboratory of Vegetation Restoration and Management of Degraded Ecosystems, South China Botanical Garden, Chinese Academy of Sciences, Guangzhou 510650, China
*   Correspondence: dengqi@scbg.ac.cn (Q.D.); wuwencheng@scies.org (W.W.); Tel.: +86-20-8411-0052 (Q.D.); +86-020-8557-4549 (W.W.)

**Abstract:** Biochar addition has been proposed to influence soil nitrogen (N) cycle and improve crop productivity. However, a comprehensive understanding of the impact of soil N cycle on the productivity of different crops under biochar addition remains elusive. Thus, a meta-analysis of 93 peer-reviewed field experiments was undertaken to investigate these outcomes of biochar addition. Results show that biochar addition significantly enhances crop productivity by 13.0%. The productivities of legumes, maize, and wheat were significantly increased by 21.2%, 14.3%, and 8.00% following biochar addition in the fields, respectively. However, the improvement in rice productivity is the lowest (3.36%), insignificant following biochar addition. The aggregated boosted tree, and partial least squares path analyses, indicated that the changes in the soil N pool (i.e., TN, $NO_3^-$-N, and $NH_4^+$-N) and plant N uptake were the most critical factors in increasing crop productivity under biochar addition. Although biochar addition had no significant enhancement on rice productivity, enhancing field rice nitrogen uptake and modest application of nitrogen fertilizers greatly improved rice productivity. The amount of soil $NH_4^+$-N was vital to improving legume productivity rather than biological $N_2$ fixation when biochar was applied. Increases in $NH_4^+$-N content and decreasing $NO_3^-$-N content were favorable to improving maize productivity under biochar addition. In contrast, biochar additions did not significantly regulate the parameters and processes of soil N cycle to enhance wheat productivity. Overall, the productivity of different types of crops is greatly influenced by soil N cycle under biochar addition.

**Keywords:** biochar addition; crop productivity; soil N cycle; field studies; meta-analysis

## 1. Introduction

Nitrogen (N) is required by plants in considerable quantities and is the most frequent limiting factor for crop biomass and yield (productivity) [1]. Crop productivity is undoubtedly affected by soil N cycle, including the size and flux of N pool, the processes of N transmission (i.e., nitrification, denitrification, and N mineralization), N fixation and loss [2,3]. Notably, improving soil N availability and use-efficiency are directly linked to enhancing crop growth [4,5]. Low N use efficiency is the primary difficulty with crop N demand, surplus fertilization, a low capacity for N retention in agricultural soils, and challenges regarding the time and magnitude of soil N mineralization [6–8]. However, the addition of biochar during N fertilizer application in agricultural soils has been proposed for improving temporal synchrony between crop N demand and soil N availability

and reducing N loss (i.e., N leaching, $N_2O$ emission, and $NH_3$ volatilization) [9–11]; thus, enhancing crop productivity [12,13].

Biochar is a carbon-rich material produced by heating biomass such as wood, manure, or crop residues in an oxygen-limited environment, and is also an important carbon source [14,15]. Especially, biochar applications significantly impact on soil carbon cycle, soil microorganism abundance/activity, and plant respiration, and thus affect crop productivity [14,16,17]. Several published reports indicated that biochar additions greatly improve crop productivity [16–20]. For example, Bai et al. (2022) indicated that biochar combined with inorganic and organic fertilizers significantly increased crop yield by 25.3% and 179.6%, respectively. In contrast, negative or no effect was reported on grain yield and biomass under biochar addition without applying N fertilizers [21]. The high carbon content of biochar may cause immobilization of soil N, which would adversely affect crop growth and decrease crop productivity [11,22]. A recent study indicated that soil N cycle might be one of the most relevant outcomes of biochar on crop productivity and sustainability of agricultural systems [23–26]. Another explanation for these different results could be attributed to the different responses and utilization forms of different crops (i.e., rice, maize, wheat, and legumes) to N form in soil under the influence of biochar addition [18,19,27]. For instance, the low content of $NH_4^+$-N leads to an enhanced increase in wheat aboveground biomass [28], but a decrease in maize aboveground biomass under biochar addition [29]. Other studies also found that a high content of $NO_3^-$-N and N uptake with biochar addition could improve maize yield [30], while reducing the rice yield [31]. Therefore, it is necessary to systematically evaluate soil N cycle affecting the productivity of different crops under biochar addition. In particular, the main regulatory factors mediating soil N cycle parameters' effect on the productivity of different crops still need to be understood.

The reliability of meta-analysis is highly dependent on the quantity and quality of the data analyzed [32,33]. However, previous meta-analyses used data mainly drawn from the composite data of laboratory and field experiments, of which there are fewer, resulting in excessive, inconsistent results [18,19]. With the rapid increase in the related field studies of biochar experiments in the recent [34–36], a meta-analysis using the data collected from 93 peer-reviewed global field experiments should comprehensively analyze soil N cycle influence on the productivity of various crops under biochar addition. Consequently, the objectives of this study are (1) to quantify the effects of biochar addition on the productivity of various crops and (2) to identify the primary control factor in soil N cycle affecting the productivity of different crops under biochar addition.

## 2. Methodology

### 2.1. Data Sources and Compilation

The published articles were searched on the Web of Science and Google Scholar using the following keywords "biochar OR black carbon AND nitrogen OR N OR nitrate OR ammonium OR mineral N AND crop yield OR crop productivity AND soil AND field". Relevant articles were selected for this meta-analysis if they satisfied the following criteria: (1) only field experiments were included; (2) biochars were produced by anaerobically pyrolyzing organic materials; (3) results of N cycle parameters and the biomass and yield of different crops were concurrently reported in each paper; (4) each treatment included at least three replicates; (5) the control and biochar treatments were subject to the same management practices (i.e., tillage, irrigation, fertilization, and residue additions); and (6) the original data could be extracted from the manuscript, including the mean and standard deviation (SD) or standard error (SE). For those studies devoid of any information on the variance (SD or SE), 1/10 of the mean was used as SD in these cases [37]. The relevant databased information is included in the Supplementary Materials.

This study focuses on evaluating the effects of soil N cycle on the productivity of different crops under biochar addition. The productivity of different crops is generally represented by the values of crop total biomass, aboveground biomass, belowground

biomass, and crop yield [16]. The N cycle includes soil N pool (soil total N (TN), microbial biomass N (MBN), inorganic N (IN), $NH_4^+$-N, and $NO_3^-$-N), N fixation (biological $N_2$ fixation (BNF)), plant N uptake (PNU), N loss ($NH_3$ volatilization ($NH_3V$), $N_2O$ emission ($N_2OE$), and N leaching (NL)), and N transformation. N transformation was indicated by the microbial indicators, including soil microbial abundance (SMA), the abundance of nitrifying (amoA) and denitrifying genes (DENG) (including amoA (archaeal ammonia (AOA) and bacterial ammonia (AOB) oxidizers), nitrate reductase gene (narG), nitrite reductase genes (nirK/S), and nitrous oxide reductase gene (nosZ)) [38,39]. Crop type categories extracted from the papers mainly included rice, maize, wheat (containing wheat and barley), legumes (i.e., peanut, cowpea, and bean), forage grass, cotton, tuber (e.g., potato, and onions), vegetables (such as arugula, cabbage, lettuce, canola, broccoli, and coriander herb)), and sugarcane [19].

### 2.2. Data Acquisition and Analysis

The raw data were obtained numerically from text and tables or extracted from the figures in the original papers using the Get-Data Graph Digitizer 2.26. The effects of biochar on the different sampling times and the uppermost soil layer were chosen [14,38]. SE values were unified into SD values (SD = SE $\times \sqrt{n}$, where n is the replicate number) [40].

The effects of biochar addition on N cycle parameters and the productivity of different crops were evaluated using the response ratio (RR) [37]:

$$RR = \ln\left(\frac{X_t}{X_c}\right) \tag{1}$$

where $X_t$ and $X_c$ are the results of the biochar treatment and control treatment, respectively. Moreover, SD and the number of replicates were used as a measure of variance. The weight for each effect size was considered as its inverse variance. The effect sizes of the above-categorized groups were calculated using a categorical random-effects model [14,41].

Mean effect sizes of each category and the 95% confidence intervals (CIs) generated by bootstrapping (999 iterations) were calculated using the "metafor" packages in R software (version 4.0.5, R, Hongkong). The effect sizes (RR) were converted to percentage change, following methods outlined in a previous report [41]. Mean effect sizes were considered significantly different from zero if the 95% confidence intervals (CIs) do not overlap zero and considered significantly different from each other if their 95% CIs do not overlap [37,38]. The mean of all effect sizes combined was calculated for soil N cycle to the productivity of various crops under biochar addition in the conditions of global field.

To further evaluate the related data validity and the robustness of this meta-analysis, the fail-safe number and funnel plot were used to elucidate the publication bias [42,43], which was compared with $5n + 1$ ($n$ is the number of cases). The results of different crop productivity and N cycle parameters from the datasets were without publication bias (Tables S1–S3 and Figure S1). The between-group heterogeneities ($Q_B$) were calculated across all datasets for a given response variable (i.e., Figure 1), and the chi-square test was applied to determine the significant difference between groups at $p < 0.05$ [38].

The quantification of mainly controlled factors (the percentage of relative influence) in the different conditions was elucidated using an aggregated boosted tree (ABT) model. The "gbm" package in R software (version 4.0.5, R, Hongkong) was applied. Model selection was run with the following predictors: TN, MBN, IN, $NH_4^+$-N, $NO_3^-$-N, BNF, PNU, $NH_3V$, $N_2OE$, NL, SMA, amoA, and DENG. The partial least squares (PLS) path analysis (a structural equation model) differed from the conventional covariance-based path analysis, and did not impose distributional assumptions on the data, which was usually difficult to meet [44]. Therefore, the criteria used in the covariance-based approaches are invalid for the PLS path analysis. In the PLS path analysis, the loading of each indicator variable was the key to estimating latent variable scores and calculated as the correlation between a latent variable and its indicators. An iterative algorithm was used to estimate the loadings until convergence was reached to maximize the explained variance of the

dependent variables (both latent and observed indicator variables). A non-parametric bootstrapping (200 resamples in this study) was used to estimate the precision of the PLS parameter estimates. The 95% bootstrap CIs were used to judge whether the estimated path coefficients were significant. The PLS path analyses were performed using R software's (version 4.0.5, R, Hongkong) "plspm" packages.

## 3. Results

### 3.1. Soil N Cycle Influences the Productivity of Different Crops under Biochar Addition

Under biochar addition, crop productivity was overall enhanced by 13.0% in the global field conditions (Figure 1). Tuber productivity greatly increased most by 24.5%, while rice productivity was the lowest (3.36%) without significant increase under biochar addition (Figure 1). The productivities of legumes, vegetables, maize, forage grass, and wheat were significantly increased by 21.2%, 15.8%, 14.3%, 9.55%, and 8.00% under biochar addition in the field conditions, respectively (Figure 1). Moreover, cotton productivity was also enhanced by 7.90%, albeit insignificant under biochar addition (Figure 1).

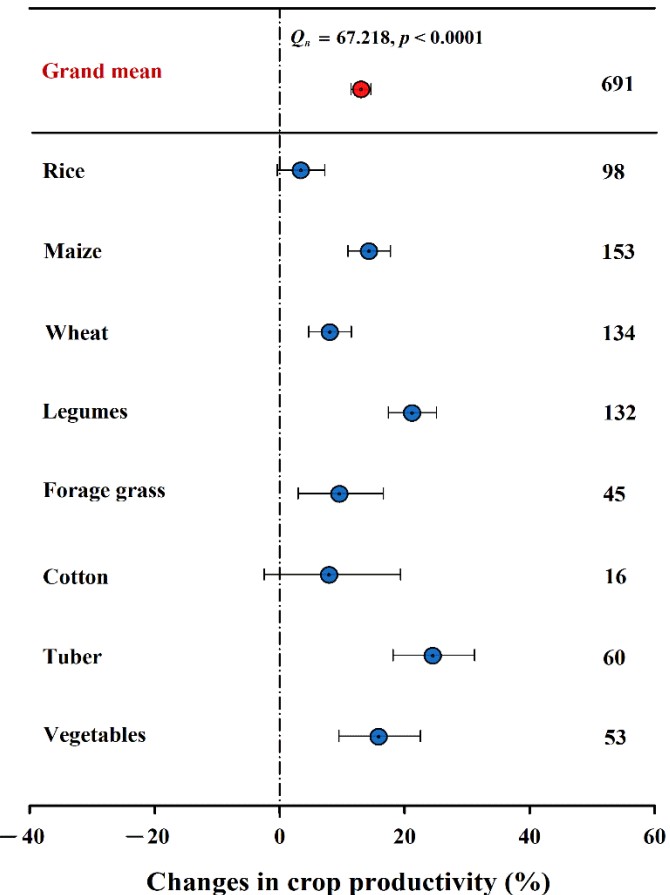

**Figure 1.** Effects of biochar addition on the productivity of various crops. Data on the right side of each panel represent the number of observations. Bars indicate 95% confidence intervals.

The data shows that only the RR of TN was positively correlated with the RR of wheat productivity ($p < 0.01$) under biochar addition (Figure 2a and Table 1). The RR of rice productivity was positively correlated with the RRs of TN ($p < 0.05$) and PNU ($p < 0.001$), and negatively correlated with the RRs of $NO_3^--N$ ($p < 0.05$), $NH_4^+-N$ ($p < 0.001$), and $N_2O$ emissions under biochar addition (Figure 2b and Table 1). The RR of legume productivity had a positive correlation with the RR of PNU ($p < 0.01$; Table 1), but a negative correlation with the RR of biological $N_2$ fixation by biochar addition (Figure 2d). A positive association was seen between the RRs of maize productivity and PNU ($p < 0.01$) under biochar addition

(Figure 2c). The RR of maize productivity was positively correlated with the RR of $NH_4^+$-N ($p < 0.05$) under biochar addition (Table 1). Moreover, biochar addition markedly increased the N uptake of maize and legumes (Table 2).

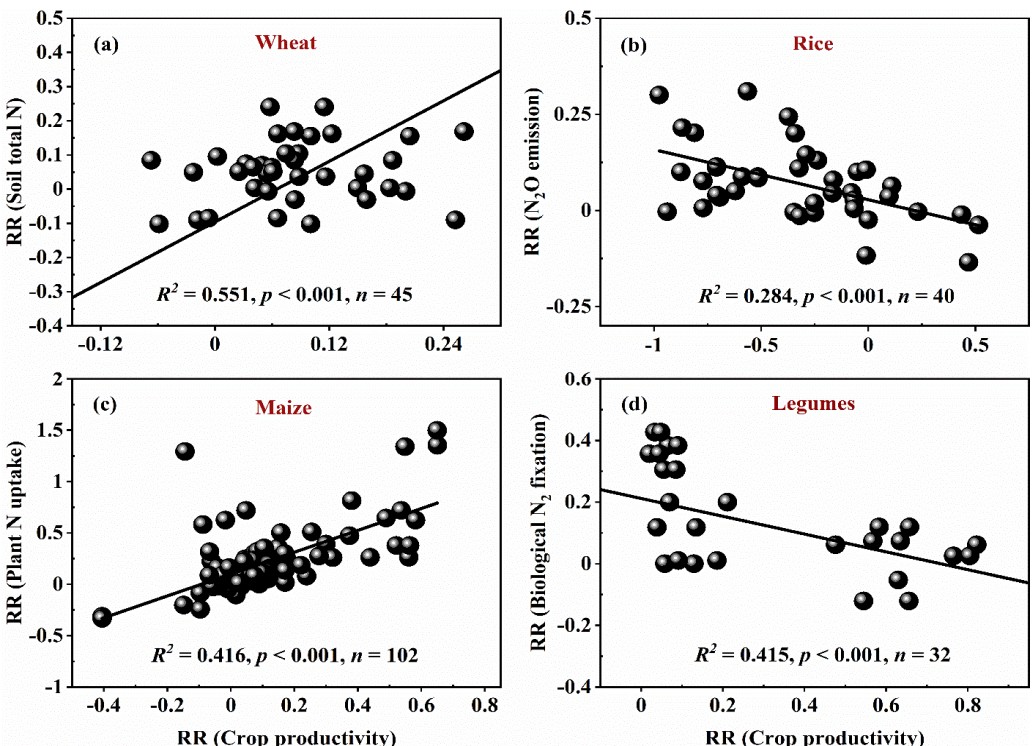

**Figure 2.** Relationships between the response ratios of wheat, rice, maize, and legume productivity vs. the response ratios of (**a**) soil total N, (**b**) $N_2O$ emission, (**c**) plant N uptake, and (**d**) biological $N_2$ fixation.

**Table 1.** Pearson correlation coefficients ($r$) between the response ratios (Equation (1)) of different crops' productivity and soil nitrogen (N) cycle parameters. Sample size is depicted by $n$.

| Index | $r$ | $p$ Value | $n$ |
|---|---|---|---|
| | | Rice | |
| TN | 0.256 | 0.047 | 60 |
| $NO_3^-$-N | −0.382 | 0.018 * | 38 |
| $NH_4^+$-N | −0.543 | 0.000 *** | 44 |
| PNU | 0.745 | 0.000 *** | 29 |
| | | Maize | |
| TN | 0.073 | 0.566 | 65 |
| $NO_3^-$-N | −0.369 | 0.005 ** | 35 |
| $NH_4^+$-N | 0.339 | 0.046 * | 57 |
| $N_2O$ emission | −0.174 | 0.200 | 55 |
| | | Wheat | |
| $NO_3^-$-N | 0.113 | 0.490 | 40 |
| $NH_4^+$-N | 0.190 | 0.282 | 34 |
| $N_2O$ emission | 0.037 | 0.838 | 33 |
| PNU | 0.215 | 0.292 | 26 |
| | | Legumes | |
| TN | 0.211 | 0.059 | 81 |
| $NO_3^-$-N | −0.061 | 0.704 | 41 |
| $NH_4^+$-N | 0.177 | 0.269 | 41 |
| $N_2O$ emission | −0.549 | 0.065 | 12 |
| PNU | 0.440 | 0.005 ** | 40 |

Note: *, **, and *** represent the significance levels of $p < 0.05$, $p < 0.01$, and $p < 0.001$, respectively. TN and PNU represent soil total N and plant N uptake, respectively.

**Table 2.** Mean values and 95% confidence interval (CI) of percentage changes ($P_c$: %) in soil N cycle parameters under biochar addition in various crops.

| Index | Effect Size | Rice | Maize | Legumes | Wheat |
|---|---|---|---|---|---|
| TN | $P_c$ (n) | 9.72 (43) | 10.42 (50) | 12.82 (39) | 4.29 (24) |
| | 95% CI | −0.25~20.7 | 1.05~20.7 | 1.99~24.8 | −8.45~18.5 |
| $NO_3^- $-N | $P_c$ (n) | −4.22 (25) | 1.17 (34) | 15.12 (19) | 2.68 (25) |
| | 95% CI | −11.1~3.24 | −5.03~7.77 | 6.90~24.0 | −4.96~10.9 |
| $NH_4^+ $-N | $P_c$ (n) | 17.38 (29) | 2.81 (21) | 18.72 (19) | −8.68 (21) |
| | 95% CI | 10.3~24.9 | −3.94~10.0 | 8.82~29.5 | −14.7~−2.23 |
| PNU | $P_c$ (n) | 0.37 (23) | 27.07 (67) | 36.51 (20) | 10.84 (15) |
| | 95% CI | −8.52~10.1 | 20.3~34.3 | 22.9~51.6 | −1.94~25.3 |

Note: TN and PNU represent soil total N and plant N uptake, respectively. n is the sample size.

### 3.2. Major Control Factors in Soil N Cycle Effect on the Productivity of Various Crops

The ABT analysis was carried out to compare the relative importance (major influence factors) of N cycle parameters with biochar addition on overall crop productivity (Figure 3). In total, 92.8% of the variances in the crop productivity are explained by the first six factors of TN, MBN, $NH_4^+$-N, $NO_3^-$-N, BNF, and PNU. Notably, the factors of soil N pool (i.e., TN, $NH_4^+$-N, and $NO_3^-$-N) and N fixation (i.e., PNU) were the most influential variables on crop productivity under biochar addition among the 13 chosen variables (Figure 3).

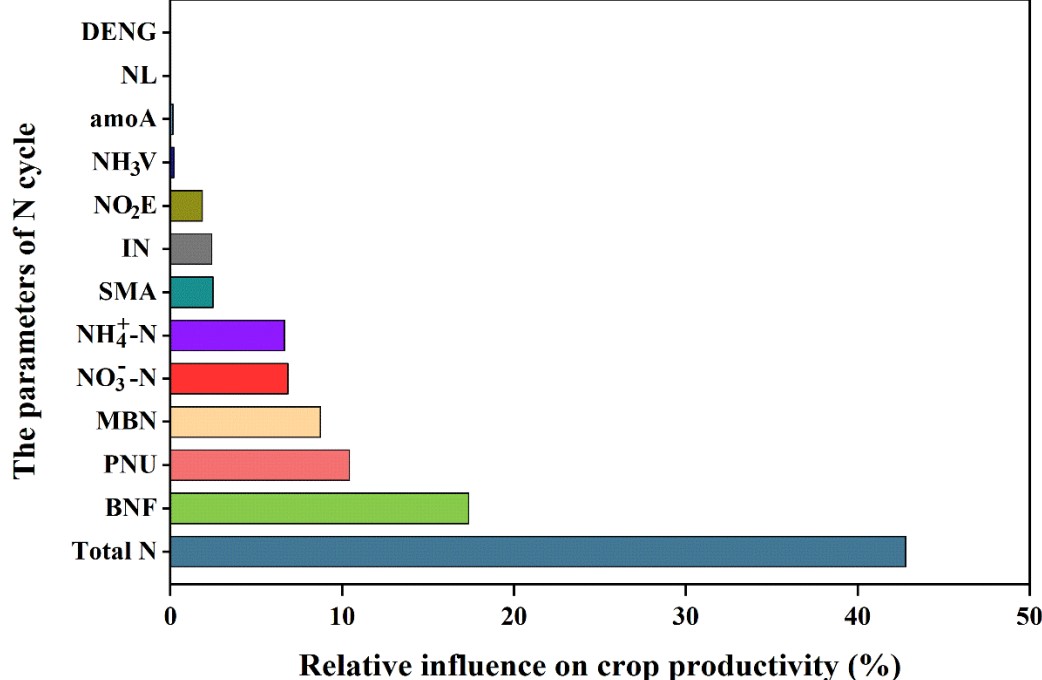

**Figure 3.** The relative influence (%) of the effects of soil N cycle (microbial biomass N: MBN; inorganic N: IN; ammonium N: $NH_4^+$-N; nitrate N: $NO_3^-$-N; $NH_3$ volatilization: $NH_3$V; $N_2O$ emission: $N_2OE$; N leaching: NL; biological $N_2$ fixation: BNF; plant N uptake: PNU; soil microbial abundance: SMA; archaeal ammonia and bacterial ammonia oxidizers: amoA; denitrification genes: DENG (nitrate reductase (narG), nitrite reductase (nirK/S) and nitrous oxide reductase (nosZ) genes)) following biochar addition on crop productivity based on aggregated boosted tree (ABT) model analysis.

Soil N pool and N fixation acted as the primary influence factors in crop productivity under biochar addition by the ABT analysis (Figure 3). On this basis, PLS path analysis was used to further explore the responses of soil N pool (i.e., TN) and N fixation (i.e., PNU) to the productivity of different crops with biochar addition. The results showed that the goodness of fit indicating the average prediction of the entire model was 0.27–0.63 for

the path analysis of different crop productivity (Figure 4). Under biochar addition, rice productivity was not significantly associated with soil N pool, but directly and positively correlated with PNU ($p < 0.05$; Figure 4). Soil N pool (i.e., TN, NH$_4^+$-N, and NO$_3^-$-N) had a direct and positive correlation with legume productivity ($p < 0.01$), but negative associations with maize productivity ($p < 0.05$; Figure 4). The PLS path analysis also indicated that soil NH$_4^+$-N and NO$_3^-$-N were the most significant factors influencing legume and maize productivity (Figure 4). Moreover, the improvements in TN, NH$_4^+$-N, and NO$_3^-$-N contents greatly enhanced legume productivity (Table 2).

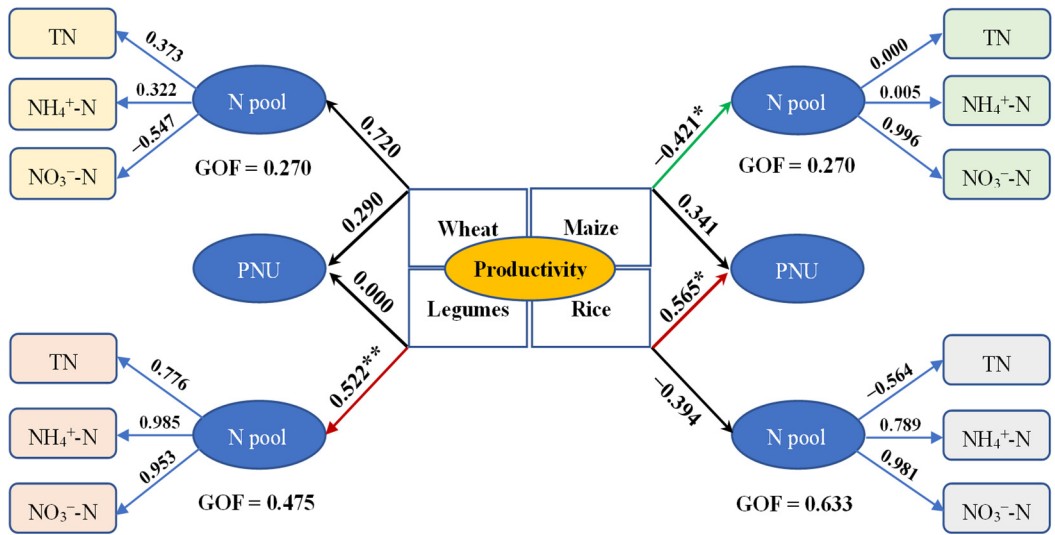

**Figure 4.** Partial least squares (PLS) path analysis results on the direct and indirect effects of soil N pool (soil total N: TN; ammonium N: NH$_4^+$-N; and nitrate N: NO$_3^-$-N) and plant N uptake (PNU) on wheat, maize, rice, and legume productivity. Numbers show the path coefficients. Green and solid red paths indicate negative and positive effects (* and ** represent $p < 0.05$ and 0.01); The PLS path analysis was assessed using the Goodness of Fit statistic (GOF), and the GOF value was 0.270-0.633. the corresponding latent variables with their considered indicators are under the path analysis results. Numbers below the index show the loading scores (the correlation coefficient between the latent variable and its indicators).

## 4. Discussion

### 4.1. Effect of Biochar Addition on the Productivity of Different Crops

With biochar addition, tubers show the highest increase by 24.5% in productivity among the different crops (Figure 1). This could be attributable to the improved soil pH, water availability, texture, and nutrients by biochar addition, which were more beneficial for aboveground and belowground growth of tuber than other crops (e.g., maize and wheat) [14,45,46]. On the other hand, biochar addition can reduce soil phytotoxic compounds and pathogens; thus, increase tuber productivity [47,48]. Similarly, the productivities of maize, wheat, legumes, forage grass, and vegetables were significantly increased under biochar additions in the fields (Figure 1). This could also be due to the liming, structural, and nutrient effects of biochars, improving crop productivity [18,19,49]. Furthermore, biochar addition greatly improved plant N uptake and use efficiency, and reduced N leaching and greenhouse gas emission, and thus significantly increased crop productivity [19,38,49]. These results were consistent with previous meta-analysis reports that biochar addition positively impacted the productivity of different crops in the global field conditions [18,19].

However, the current data indicated that rice productivity was not significantly enhanced by biochar addition in the field conditions (Figure 1). This was consistent with the findings of Liu et al. (2013, 2019), and Ye et al. (2019), which reported that rice productivity

was less responsive to biochar than the productivity of dry upland crops. The possible reason was that the high C/N ratio of biochar addition could influence N utilization or limit rice rooting and growth; thus, decrease rice productivity [34,50,51]. Alternately, the rice field with higher soil moisture than in drylands could weaken the liming and structure effects of biochar; thus, lead to the non-significant improvement in rice productivity with biochar addition [19,52]. The current study also indicated that rice productivity decreased with increasing soil $NH_4^+$-N and $NO_3^-$-N contents under biochar addition (Table 1). However, a previous study found that biochar addition significantly enhanced soil $NH_4^+$-N and $NO_3^-$-N contents in the field conditions [38]. Therefore, the inference was that the high levels of applied N fertilizers should be reduced under biochar addition in the paddy soil.

### 4.2. Soil N Cycle Influences the Productivities of Maize and Legumes with Biochar Addition

Previous meta-analyses generally demonstrated that productivity (yield) of maize showed an enhanced response to biochar addition with and without fertilizer (the control treatment is non-fertilized) than those of other major dryland cereals (i.e., wheat, barley, and oat) [19]. The current study indicated that the enhancement of maize productivity by biochar addition was positively correlated to higher amounts of soil $NH_4^+$-N content and plant N uptake, and negatively to soil $NO_3^-$-N content (Figure 2c, Tables 1 and 2). The PLS path analysis also showed that $NO_3^-$-N content in the soil had the most significant adverse effect on maize productivity (Figure 4). However, plant N uptake had no significant impact on maize productivity under biochar addition (Figure 4). These results imply that $NO_3^-$-N and $NH_4^+$-N in the soil N pools could be the more critical factors for maize productivity than plant N uptake. Notably, the increased $NH_4^+$-N content and decreased $NO_3^-$-N content are the most advantageous factors for improving maize productivity under biochar addition in the global field conditions (Table 1). Moreover, previous studies found that the application of biochar combined with N fertilizer (i.e., N application rates of 156–170 kg N ha$^{-1}$) increasing the stock of $NH_4^+$-N and decreasing $NO_3^-$-N content in the field soil profile had a positive effect on maize yield [53,54].

Biochar addition increased the productivity of legumes more remarkably than main cereals (such as maize) in the dryland soil (Figure 1), which was consistent with previous results [18,19,55]. These studies demonstrated that biochar could specifically enhance biological $N_2$ fixation in legumes, and then promote legume growth more than other crops (i.e., maize and wheat) without fixing atmospheric $N_2$ [19,56,57]. Nevertheless, this study revealed that the improved biological $N_2$ fixation was not beneficial to increasing legume productivity under biochar addition (Figure 2d). A possible explanation was that the reduction of soil $NH_4^+$-N and $NO_3^-$-N contents enhanced the percentage of biological $N_2$ fixation with biochar addition [58]. However, biochar addition significantly increased soil mineral N in the field conditions [38], which directly supplied nutrients to increase legume growth, and thus could inhibit the nutrient source of biological $N_2$ fixation. A high N uptake could decrease the amount of biological $N_2$ fixation by reducing the mycorrhizae of legumes in the biochar treatment (Tables 1 and 2) [17,59].

On the other hand, plant N uptake exhibited an insignificant correlation with legume productivity, while soil N pool notably influenced legume productivity under biochar addition. Especially, $NH_4^+$-N content showed the most positive correlation with legume productivity (Figures 3 and 4). Meanwhile, biochar addition greatly improved soil $NH_4^+$-N in legumes (Table 1). Therefore, these results suggested that the amount of soil $NH_4^+$-N could collectively regulate legume productivity under biochar addition.

### 4.3. Responses of Wheat and Rice Productivities to Soil N Cycle with Biochar Addition

Only soil TN in the N cycle was positively correlated with wheat productivity (Figure 2a; Table 1), while soil N pool and N uptake did not significantly affect wheat productivity under biochar addition in the current study (Figure 4). Meanwhile, biochar addition did not increase the amount of soil N contents and N uptake in the wheat fields (Table 2). These results suggested that soil N cycle could not significantly and directly

influence wheat productivity under biochar addition. A previous study also indicated that wheat plants only utilized partly mineral N (i.e., $NO_3^-$-N) under biochar addition and thus could decrease the effect of soil N cycle on wheat productivity [60]. However, the soil N cycle's lack of influence on wheat productivity requires further clarification and research.

Although biochar addition had a less significant influence on increasing rice productivity, enhancing rice N uptake by reasonable biochar management practices and N fertilizer application could improve the situation [16,61]. The current study elucidated that plant N uptake in the N fixation was a more vital impact factor on rice productivity than soil N pool in this study (Figure 4). Previous research reported that the improvement in rice N uptake could be beneficial to enhancing rice growth and productivity under biochar addition [62]. Moreover, the high amount of $NH_4^+$-N in the soil N pool was detrimental to increasing rice productivity by biochar addition (Tables 1 and 2). However, Zhang et al. (2020) reported that rice was a typical species that preferred $NH_4^+$-N rather than $NO_3^-$-N. Such inconsistencies may be related to rice metabolism, the energy utilized for N assimilation inside the rice, the rhizosphere pH, and the possible solubilization of other minerals in the paddy soil with biochar addition [50,63]. Nonetheless, the relevant mechanism is not understood and needs further study.

## 5. Conclusions

Based on this meta-analysis using the global field studies data, biochar addition greatly enhanced several crop productivities overall. The changes in N pool (TN and $NO_3^-$-N) and N fixation (BNF and PNU) in soil N cycle processes were the most critical factors in enhancing crop productivity under biochar addition. Increasing rice N uptake was the essential factor for improving rice productivity under biochar addition. The amount of soil $NH_4^+$-N plays a vital role in improving legume productivity. The controlling soil $NO_3^-$-N and $NH_4^+$-N contents were important factors for maize productivity. However, the soil N cycle did not significantly affect wheat productivity when biochar was applied in the fields. For the practical application/values, our results suggest that (i) increasing the supply of soil $NH_4^+$-N will be conducive to improving legume productivity under biochar addition, (ii) improving rice N uptake and use efficiency should be beneficial to increasing rice productivity by biochar addition, and (iii) the increased soil $NH_4^+$-N application but decreased $NO_3^-$-N supply greatly enhance maize productivity under biochar addition in the field conditions. The soil N cycle under biochar addition in field conditions significantly influences the productivity of different crops.

**Supplementary Materials:** The following supporting information can be downloaded at: https://www.mdpi.com/article/10.3390/agronomy12081857/s1. Figure S1: Funnel plots of the effect sizes of abundance of (a) crop productivity, the productivities of (b) rice, (c) maize, (d) wheat, (e) legumes, (f) tuber, (g) vegetables, (h) forage grass, (i) cotton, and (j) inorganic N (IN) in the investigated datasets, Table S1: Values of fail-safe number of crop productivity (CP), and the productivities of rice, maize, wheat, legumes, tuber, vegetables, forage grass, and cotton in the investigated datasets, Table S2: Fail-safe number values of soil total N (TN), microbial biomass N (MBN), $NH_4^+$-N, $NO_3^-$-N, inorganic N (IN), soil microbial abundance (SMA), and amoA including archaeal ammonia (AOA) and bacterial ammonia (AOB) oxidizers in the investigated datasets, Table S3: The values of fail-safe number of soil denitrification genes (DENG: including nitrite reductase genes (nirK and nirS), nitrate reductase gene (narG), and nitrous oxide reductase gene (nosZ)), $N_2O$ emission (N$_2$OE), $NH_3$ volatilization (NH$_3$V), soil N leaching (NL), biological $N_2$ fixation (BNF), and plant N uptake (PNU) in the investigated datasets.

**Author Contributions:** Conceptualization, L.Z. (Leiyi Zhang) and W.W.; data curation, L.Z. (Leiyi Zhang) and Z.W.; formal analysis, Y.L.; funding acquisition, L.Z. (Leiyi Zhang), W.W. and Q.D.; investigation, L.Z. (Lingli Zhou) and J.Z.; methodology, L.Z. (Leiyi Zhang) and Y.X.; project administration, W.W.; supervision, W.W.; visualization, L.Z. (Leiyi Zhang); writing—original draft, L.Z. (Leiyi Zhang); writing—review and editing, W.W., R.Z. and Q.D. All authors have read and agreed to the published version of the manuscript.

**Funding:** This study was jointly supported by grants from the Fundamental Research Funds for the National Natural Science Foundation of China (32101397, 31870461), the Guangdong Basic and Applied Basic Research Foundation (2021A1515011559), the Research Fund Program of Guangdong Provincial Key Laboratory of Environmental Pollution Control and Remediation Technology (2020B1212060022), and the Fundamental Research Funds for the Central Public Welfare Scientific Institutes (PM-zx703-202105-179).

**Institutional Review Board Statement:** Not applicable.

**Informed Consent Statement:** Not applicable.

**Data Availability Statement:** The authors confirm that the data supporting the findings of this study is available within the article and its Supplementary Materials.

**Conflicts of Interest:** The authors declare that they have no known competing financial interest or personal relationships that could have appeared to influence the work reported in this paper.

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
