# Peer review of "Meta-Analysis of the Response of the Productivity of Different Crops to Parameters and Processes in Soil Nitrogen Cycle under Biochar Addition"

_agronomy, doi:10.3390/agronomy12081857_

Round 1
Reviewer 1 Report
In this manuscript, authors investigated the effects of the biochar addition on crop production worldwide with a meta-analysis of 93 peer-re-25 viewed field experiments. The paper was drafted properly; however, there still exits some grammar errors, such as “Moreover, previous studies also found that” (abstract part). Please check through the text and revise all of these. Just one concern about this study, would the authors be able to tell us what is the practical information/values could be applied to the real crop production from this paper?
Author Response
Dear Reviewer:
The revised manuscript is submitted. We greatly appreciate the valuable comments from you, which indeed help to improve the quality of our paper. Following your comments and suggestions, we have revised the manuscript. An itemized response to the comments was summarized in the attachment. Please check it.
Thank you for your suggestions. Best regards.
Sincerely yours,
Wencheng Wu, Ph.D., Professor
CC : Leiyi Zhang, Zhuohao Wu, Jingyan Zhou, Lingli Zhou, Yang Lu, Yangzhou Xiang, Renduo Zhang, Qi Deng

Reviewer 2 Report
Biochar has great application value in agriculture. The impact of biochar on crop productivity and its mechanism need to be comprehensively studies. Thus, the research content is of great importance. In view of the current draft, the following questions are raised.
1. Biochar is an important carbon source, and its impact on soil carbon cycle, soil microorganisms and plant respiration will also affect crop productivity. This is not mentioned in this draft. It is suggested that the role of biochar in these aspects should be described in this paper.
2. The necessary instructions about calculation of crop productivity should be available in the section of methodology.
3. The abundance of nitrifying (amoA) and denitrifying genes was involved in this draft. What can be explained by the results of these parameters?
4. It was mentioned in the draft that biochar could improve the productivity of most plants, however, it had no promotion effect on the yield of some of the plants. How to view the application prospect of biochar? It is suggested to mention it in the discussion.
Author Response

(The authors gave the same response as above.)
